# Managing Bladder Cancer Care during the COVID-19 Pandemic Using a Team-Based Approach

**DOI:** 10.3390/jcm9051574

**Published:** 2020-05-22

**Authors:** Tina Wang, Sariah Liu, Thomas Joseph, Yung Lyou

**Affiliations:** Department of Medical Oncology & Experimental Therapeutics, City of Hope Comprehensive Cancer Center, Duarte, CA 91010, USA; tinawang@coh.org (T.W.); sarliu@coh.org (S.L.); thojoseph@coh.org (T.J.)

**Keywords:** bladder cancer, urothelial carcinoma, COVID-19, team-based medicine

## Abstract

The recent novel coronavirus, named coronavirus disease 2019 (COVID-19), has developed into an international pandemic affecting millions of individuals with hundreds of thousands of deaths worldwide. The highly infectious nature and widespread prevalence of this disease create a new set of obstacles for the bladder cancer community in both delivering and receiving care. In this manuscript, we address the unique issues regarding treatment prioritization for the patient with bladder cancer and how we at City of Hope have adjusted our clinical practices using a team-based approach that utilizes shared decision making with all stakeholders (physicians, patients, caregivers) to optimize outcomes during this difficult time. In addition to taking standard precautions for minimizing COVID-19 risk of exposure for those entering a healthcare facility (screening all personnel upon entry and donning facemasks at all times), we suggest the following three measures: (1) delay post-treatment surveillance visits until there is a decrease in local COVID-19 cases, (2) continue curative intent treatments for localized bladder cancer with COVID-19 precautions (i.e., choosing gemcitabine/cisplatin (GC) over dose-dense methotrexate, vinblastine, doxorubicin, cisplatin (ddMVAC) neoadjuvant chemotherapy), and (3) increase the off-treatment period between cycles of palliative systemic therapy in metastatic urothelial carcinoma patients.

## 1. Introduction

Recently, a novel coronavirus, named severe acute respiratory syndrome coronavirus 2 (SARS-CoV-2), has developed into an international pandemic affecting millions of individuals in more than 150 countries with hundreds of thousands of deaths worldwide [1,2]. This disease has been named coronavirus disease 2019 (COVID-19) by the World Health Organization (WHO) [1]. Patients with this disease are at high risk for developing septic shock and hypoxemia, which can frequently progress to acute respiratory distress syndrome (ARDS) and death [3]. This disease creates a new set of obstacles for the bladder cancer community in both delivering and receiving care. In this manuscript, we address the unique issues regarding treatment prioritization for the patient with bladder cancer and how we at City of Hope have adjusted our clinical practices using a team-based, shared decision approach with all stakeholders (patients, caregivers, and physicians) to optimize outcomes during this difficult time. 

## 2. Balancing the Need for Bladder Cancer Treatments and Risk of Exposure to COVID-19

### 2.1. Patients with Bladder Cancer Undergoing Treatments Are at a Higher Risk for COVID-19 Infections and Worse Outcomes Compared to the General Population without Cancer 

For the patient with bladder cancer undergoing treatment, there are several safety issues that place them at higher risk of infection for COVID-19 compared to the general population without cancer. First, patients must physically leave the safety of their residences to go to the clinic, infusion center, or imaging facility where they could potentially be exposed to COVID-19. Second, the platinum-based chemotherapy regimens commonly used in bladder cancer treatments are immunosuppressive and place them at a higher risk for infection. Third, many bladder cancer patients tend to be of older age and also have multiple medical comorbidities, which has been shown to place them in a group with worse outcomes for COVID-19 [2,4]. A retrospective study that examined the outcomes of approximately 72,000 patients with COVID-19 found that those with older age and presence of medical comorbidities were associated with adverse outcomes [2,4]. In another retrospective study by Liang and colleagues, it was suggested that patients with a history of cancer itself may be associated with worse outcomes from COVID-19 [5,6]. However, it should be noted that this particular retrospective study was limited in that only 18 of the 1590 patients who were studied had a history of cancer, making it difficult to form a general conclusion from such a small sample size [5,6]. Regardless, based on the other reasons discussed above, it is clear that patients with bladder cancer undergoing active therapy or post-treatment surveillance are at a higher risk for COVID-19 exposure and could potentially suffer worse outcomes compared to the general population.

### 2.2. Prioritizing Treatments Appropriately and Applying Social Distancing

Ensuring patient safety is the key principle when it comes to delivering medical care among all healthcare professions. In the setting of the COVID-19 pandemic, the central question we have asked ourselves as providers while managing each patient’s care has been: Will delaying the patient’s bladder cancer treatment in accordance with current COVID-19 social distancing measures lead to a worse long-term outcome? Current models suggest that this pandemic may proceed until herd immunity or a vaccine is developed, with repeated waves of infections, which some experts estimate could continue for another 18 months. Since it is not feasible to delay bladder cancer treatments for another 18 months, we at City of Hope have developed a consensus framework to help balance these competing risks (Figure 1). By utilizing this framework, we have been able to guide our clinicians within the network on how to make a shared decision with the patient that can prioritize bladder cancer treatments appropriately while minimizing the risk for COVID-19 exposure (Figure 1). 

### 2.3. Applying COVID-19 Risk Mitigation Measures for Bladder Cancer Treatment

In the state of California, there is a “shelter in place” order that was initiated on 19 March 2020 along with other social distancing measures due to the concern that individuals may be at high risk of becoming infected and could also infect others, further propagating this pandemic. Current epidemiology modeling suggests that the peak incidence of COVID-19 will have occurred sometime in mid-to-late April in the state of California. This framework assumes that the number of new cases will start to decrease in the months of May and June 2020. In the case that there is indeed a second wave of infections later during the fall and winter months of 2020, one could reapply this framework based on the expected peaks. As a result, we suggest the following framework to assist the practicing oncologist in determining optimal treatment strategies for the patient with bladder cancer. 

#### 2.3.1. Delay Post-Treatment Surveillance Visits until There Is a Decrease in COVID-19 Cases

For patients undergoing surveillance imaging after completion of cystectomy or other definitive therapies, the National Comprehensive Cancer Network (NCCN) guidelines currently recommend imaging every 6–12 months [7]. Keeping these guidelines in mind, we have rescheduled the patient’s clinic and imaging visit to avoid the expected COVID-19 peak period (April–May) so that it will take place during the next 2–3 months in June or July as a way to minimize risk of exposure. 

#### 2.3.2. Continue Curative Intent Treatments for Localized Bladder Cancer with COVID-19 Precautions 

Even in these difficult times, urothelial bladder cancer is an aggressive disease with poor prognosis when it progresses to metastatic disease. Therefore, we have been vigilant in continuing to deliver curative intent treatments when patients have localized urothelial carcinoma, if possible in a timely manner. A meta-analysis of 13 studies suggested that a delay of more than 12 weeks from time of diagnosis to execution of radical cystectomy, only in muscle invasive urothelial cancer, was associated with worse outcomes [8]. Another study showed that initiating neoadjuvant chemotherapy (NAC) with a delay of more than 8 weeks from time of diagnosis led to worse outcomes [9]. Therefore, we have continued to offer cisplatin-eligible patients NAC within 8 weeks and cisplatin-ineligible patients radical cystectomy within 12 weeks from time of diagnosis, while the infusion center and operating room resources are available for those patients with localized disease since there is a limited window of curative treatment opportunity.

The first set of measures we have instituted to minimize potential risk for COVID-19 exposure for all on-site people (visitors and healthcare workers) is to create a single, separate point of entry to the active clinical areas and institute a strict policy limiting visitors to patients only. Prior to entering the clinical area, all personnel (including patients and healthcare workers) are screened for COVID-19 symptoms (i.e., cough, dyspnea, and fever) and have their temperatures measured. People determined to be asymptomatic and afebrile are then required to don a face mask and are issued an entrance band indicating they have passed screening measures for that day. If someone is found to be symptomatic, we then refer this individual to an on-site “fever clinic” staffed by designated clinical personnel who have been trained and equipped with the appropriate personal protective equipment (PPE) to perform a nasopharyngeal swab for in-house COVID-19 testing. We have also repurposed one of our hospital wards with negative pressure rooms to serve as the COVID-19 unit with its own set of designated staff to decrease exposure within the facility. In both the inpatient and outpatient areas, all people (patients and healthcare workers) are required to don a face mask at all times, which has been suggested as a way to prevent sustained exposure to COVID-19 and reduce risk for infection [10]. 

The second set of COVID-19 risk mitigation measures specifically pertain to treatments used for urothelial bladder cancer. Current NCCN guidelines recommend neoadjuvant chemotherapy in muscle invasive bladder cancer [7]. In the choice of regimen, the two most commonly used regimens are dose-dense methotrexate, vinblastine, doxorubicin, cisplatin (ddMVAC) and gemcitabine/cisplatin (GC) [7,11,12]. During this time, we have advocated for using gemcitabine/cisplatin over ddMVAC for the following reasons. Although there is some discussion suggesting that ddMVAC may have a trend towards higher efficacy, it has yet to be definitively supported in a head-to-head prospective trial and retrospective studies have shown similar amounts of efficacy between these two regimens [11,12]. In addition, ddMVAC tends to be more myelosuppressive than GC, placing patients at higher risk for infections due to the neutropenia and symptomatic anemia requiring blood transfusions, which during this time have been especially challenging due to a steep drop in blood donations [11,12]. Finally, ddMVAC is given as a 14-day cycle whereas GC is given as a 21-day cycle. The 14-day cycle of ddMVAC allows a patient to proceed sooner to radical cystectomy compared to GC, but during this time we would recommend GC because it allows the oncologist to space out the patient visits and can help adhere better to the principle of social distancing [11,12]. Another measure we have taken is to implement weekly telephone checks with patients undergoing active systemic therapy. This allows us to determine if a patient is having any significant chemotherapy-related adverse effects or other acute medical issues, for which they could potentially be treated as an outpatient before they progress to needing emergency room or acute inpatient care. For example, if a patient is experiencing significant dysuria due to a potential urinary tract infection, one can prescribe antibiotics empirically at their local pharmacy and help them avoid the need to seek emergency room care, which is most likely to be overcrowded during this pandemic. For those patients that are undergoing concurrent radiation and chemotherapy with curative intent, we have continued their treatments while taking the abovementioned general COVID-19 precautions (i.e., screening at entry, donning facemasks, and weekly telephone checks). 

#### 2.3.3. Increasing Off-Treatment Period between Cycles for Palliative Systemic Therapy in Metastatic Urothelial Carcinoma Patients

For those patients already undergoing palliative first-line systemic therapy, we have continued their treatments as those regimens provide overall survival benefit. In this situation, if chemotherapy needs to be started we would recommend, as discussed above, to prescribe anti-emetics and pain medications for the patient to have immediately available at home as an outpatient. Additionally, weekly telephone checks would be conducted to prevent any chemotherapy-related complications early. Another important factor to consider, as discussed above, is lengthening the period of time between treatments. Normally, gemcitabine/cisplatin or gemcitabine/carboplatin is administered as a two weeks on, 1 week off schedule. In this case, it is reasonable to do 2 weeks on, 2 weeks off to help spread out the treatment duration as much as possible to maximize social distancing. Second-line treatment usually involves the use of immune checkpoint inhibitors such as pembrolizumab or atezolizumab. Pembrolizumab is dosed every 3 weeks, but in order to maximize social distancing for the patient, it is reasonable to stretch it to every 4 weeks during this period since it is unlikely the cancer will grow significantly during the extra week off. In this setting, atezolizumab and nivolumab already has an Federal Drug Administration (FDA)-approved every-4-week dosing, which would also make it a viable alternative. The use of third-line treatment with enfortumab vedotin requires administration once every week for 3 weeks straight and then taking the fourth week off. Again, to provide more space between visits, it would be reasonable to increase the off-treatment period from 1 week to 2 weeks to provide the patient more social distancing. 

Even during these difficult times, it is crucial to continue clinical trials to the best of our ability and help advance the field of oncology. In order to preserve needed resources for COVID-19 prevention and treatment within our institution, we have focused our efforts on continuing current open clinical trials and slowing down the pace of opening new trials. 

## 3. Conclusions

COVID-19 has developed into an international pandemic affecting millions of individuals and has created a new set of obstacles for the bladder cancer community in both delivering and receiving care. Because patients with bladder cancer require treatment even in these difficult times, we have developed a framework that utilizes a team-based approach with shared decision making among all stakeholders involved (physicians, patients, caregivers) to optimize outcomes during this difficult time. It is our hope that the conceptual framework presented above and institutional experience can be adjusted to fit the available local resources for others that are looking to balance these two competing needs when treating patients with bladder cancer during the COVID-19 pandemic. 

## Figures and Tables

**Figure 1 jcm-09-01574-f001:**
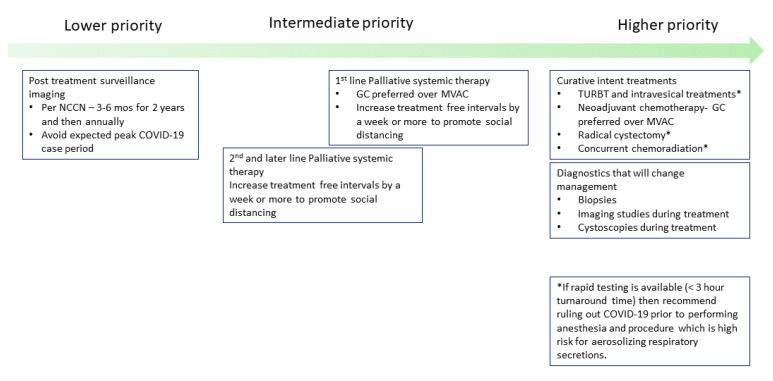
**Conceptual framework for prioritizing bladder cancer treatments during the COVID-19 pandemic.** This framework provides guidance on key treatments that should still be offered in order to ensure optimal bladder cancer outcomes if possible. We recommend that these listed priorities can be modified based on available local resources and the patient’s overall medical status.

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
