# Peer review of "Managing Bladder Cancer Care during the COVID-19 Pandemic Using a Team-Based Approach"

_jcm, 2020, doi:10.3390/jcm9051574_

Round 1

Reviewer 1 Report

Dear Editor,

thank you for the opportunity to review the manuscript (jcm-788122) devoted to bladder cancer treatment in the era of COVID19. This is a timely manuscript that should be published as it provides important clinical hints to cope with the disease. I am aware that many clinicians from all over the World should take the precautions emphasized by the authors under serious consideration. However I wish to underlie several points:

  1. Bladder cancer remains to the group of malignancies that are cigarette smoke related. Do we have any direct evidence that COVID19 may also be smoke related?
  2. Apart from precautions provided by the authors that are related to the physicians themselves any suggestions related to the hospital rearrangements? Any information on the value of reorganization of particular wards in the hospital so that they may be devoted to the status of COVID19 instead of the specific medical specialty?
  3. Minor correction in the line 162 is needed (… is would be …)

Best regards

Author Response

Dear Reviewer 1,

Thank you for your feedback. Please see our response below on your comments and how we have adjusted our manuscript to address your concerns.

1. Bladder cancer remains to the group of malignancies that are cigarette smoke related. Do we have any direct evidence that COVID19 may also be smoke related?

This is an excellent question and the current literature is controversial if smokers are at a higher risk for COVID-19 related morbidity and mortality. For example there are two recently published systematic reviews reporting both negative and positive effects of smoking on COVID-19 related outcomes.

A meta-analysis done by Farsalinos et al which pooled together data from 13 studies examining the clinical characteristics of COVID-19 patients (n=5960) hospitalized in China [1]. They authors hypothesized that because China has a high prevalence of smokers (26.6%) that there would be a higher number of smokers present amongst hospitalized COVID-19 patients if smoking led to worse [1]. To their surprise they found that the prevalence of current smokers ranged from 1.4% (95% CI 0.0-34%) to 12.6% (95% CI 10.6-14.6%) [1]. This authors reported that from a public health perspective measures should still be taken to reduce smoking but they also speculated that there may be some sort of yet to be discovered immunomodulatory effect which should be considered as a potential treatment option in COVID-19 [1].  On the other hand a separate meta-analysis done by Vardavas et al. which pooled together data from 5 studies also examining the clinical characteristics of COVID-19 patients (n=1508) in China found the opposite conclusion that smokers were more likely to have negative outcomes [2]. The authors of this meta-analysis reported that the smokers were 1.4 times more likely (RR=1.4, 95% CI: 0.98–2.00) to have severe symptoms of COVID-19 and 2.4 times more likely to require intensive care with mechanical ventilation compared to non-smokers (RR=2.4, 95% CI: 1.43–4.04) [2].Both of the above studies are limited in that they are retrospective in nature and may have not accounted for confounding factors. Hopefully with better designed studies and clinical experience the biomedical community will be able to answer the important question of smoking and its effects on COVID-19 related outcomes.

Therefore, due to the controversial nature of smoking and COVID-19 related outcomes currently being debated in the field we have intentionally chosen not to discuss this issue. This complex question is also out of the scope of the aims of this manuscript and would be better addressed as a separate article dedicated to this question.

2.  Apart from precautions provided by the authors that are related to the physicians themselves any suggestions related to the hospital rearrangements? Any information on the value of reorganization of particular wards in the hospital so that they may be devoted to the status of COVID19 instead of the specific medical specialty?

We have added the following statement (page 4, lines 127-135) to provide the above suggested information.

“If someone is found to be symptomatic we then refer those individuals to an on-site “fever clinic” staffed by designated clinical personnel who have been trained and equipped with the appropriate personal protective equipment (PPE) to perform a nasopharyngeal swab for in house COVID-19 testing. We have also repurposed one of our hospital wards with negative pressure rooms to serve as the COVID-19 unit with its own set of designated staff to decrease exposure within the facility.”

3. Minor correction in the line 162 is needed (… is would be …)

Thank you for pointing this out and we have corrected this writing error (line 172). 

Please see the attached revised manuscript with track changes and highlighting for relevant sections. Hopefully this addresses all of your concerns. Please do not hesitate to contact us if you have any further feedback. 

Bests,

Yung Lyou 

References

  1. Farsalinos, K.; Barbouni, A.; Niaura, R. Systematic review of the prevalence of current smoking among hospitalized COVID-19 patients in China: could nicotine be a therapeutic option? Intern Emerg Med 2020, doi:10.1007/s11739-020-02355-7.
  2. Vardavas, C.I.; Nikitara, K. COVID-19 and smoking: A systematic review of the evidence. Tob Induc Dis 2020, 18, doi:10.18332/tid/119324.

Reviewer 2 Report

I am glad to see the Medical community raising the question of the "price" of the unprecedented restrictive measures brought about by the COVID epidemic.  Indeed for many subsets of the population characterized by the presence of cancers or chronic diseases requiring ongoing care, the risk of COVID infection (which can be mitigated) can be far less than the risk associated with non-compliance with necessary treatment and disease management regimens.

I would like to see the manuscript expanded with a more detailed discussion of the potential health costs of delaying procedures at each priority level.  More detail would increase the utility of this manuscript to current practitioners by allowing them easy access to summary information on the costs and benefits of the contemplated treatment choice.

Author Response

Dear Reviewer 2,

Thank you for your feedback. Please see our response below on your comments and how we have adjusted our manuscript to address your concerns.

I am glad to see the Medical community raising the question of the "price" of the unprecedented restrictive measures brought about by the COVID epidemic.  Indeed for many subsets of the population characterized by the presence of cancers or chronic diseases requiring ongoing care, the risk of COVID infection (which can be mitigated) can be far less than the risk associated with non-compliance with necessary treatment and disease management regimens.

I would like to see the manuscript expanded with a more detailed discussion of the potential health costs of delaying procedures at each priority level.  More detail would increase the utility of this manuscript to current practitioners by allowing them easy access to summary information on the costs and benefits of the contemplated treatment choice.

Thank you for feedback. We have expanded the discussion by adding the following statement to provide more details on the importance for initiating definitive treatments in a timely manner (pages 6 and 7).

“Even in these difficult times because urothelial bladder cancer is an aggressive disease with poor prognosis when it progresses to metastatic disease. Therefore, we have been vigilant in continuing to deliver curative intent treatments when they have localized urothelial carcinoma if possible in a timely manner. A meta-analysis of 13 studies suggested that a delay of more than 12 weeks from time of diagnosis to execution of radical cystectomy only in muscle invasive urothelial cancer was associated with worse outcomes [1]. Another study showed that initiating neoadjuvant chemotherapy (NAC) with a delay of more than 8 weeks from time of diagnosis led to worse outcomes [2]. Therefore, we have continued to offer cisplatin eligible patients NAC within 8 weeks and cisplatin ineligible patients radical cystectomy within 12 weeks from time of diagnosis, while the infusion center and operating room resources are available in those patients with localized disease since there is a limited window of curative treatment opportunity.“

If you have any further comments or feedback please do not hesitate to contact us. 

Bests,

Yung Lyou

  1. Fahmy, N.M.; Mahmud, S.; Aprikian, A.G. Delay in the surgical treatment of bladder cancer and survival: systematic review of the literature. Eur. Urol. 2006, 50, 1176–1182, doi:10.1016/j.eururo.2006.05.046.
  2. Audenet, F.; Sfakianos, J.P.; Waingankar, N.; Ruel, N.H.; Galsky, M.D.; Yuh, B.E.; Gin, G.E. A delay ≥8 weeks to neoadjuvant chemotherapy before radical cystectomy increases the risk of upstaging. Urol. Oncol. 2019, 37, 116–122, doi:10.1016/j.urolonc.2018.11.011.